

# The effect of high-intensity plyometric training on anaerobic performance parameters: a pilot study in U17 elite A league

Mehmet Söyler[1], Raif Zileli[2], Yunus Emre Çingöz[3], Gökmen Kılınçarslan[4], İdris Kayantaş[4], Tolga Altuğ[5], Selim Asan[6], Musa Şahin[7] and Alper Cenk Gürkan[8]

[1] Vocational School of Social Sciences, Çankırı Karatekin University, Çankırı, Türkiye
[2] Faculty of Health Sciences, Bilecik Şeyh Edebali University, Bilecik, Türkiye
[3] Faculty of Sport Sciences, Bayburt University, Bayburt, Türkiye
[4] Faculty of Sport Sciences, Bingöl University, Bingöl, Türkiye
[5] Faculty of Sport Sciences, Ağrı İbrahim Çeçen University, Ağrı, Türkiye
[6] Faculty of Sport Sciences, Erzurum Technical University, Erzurum, Türkiye
[7] Faculty of Sport Sciences, Karabük University, Karabük, Türkiye
[8] Vocational School of Healthy Services, Gazi University Ankara, Ankara, Türkiye

Corresponding author
Mehmet Söyler,
mehmetsoyler@karatekin.edu.tr

## ABSTRACT

The aim of this study was to examine the effects of high-intensity plyometric training (HIPT) on some parameters in elite soccer players in the U17. Volunteer soccer players were randomly divided into two groups according to their positions: the regular training (RT) group (age: $16.71 \pm 0.47$ years; stature: $163.46 \pm 3.31$ cm; body weight: $61.04 \pm 1.59$ kg) and the HIPT group (age: $16.64 \pm 0.50$ years; stature: $165.60 \pm 3.03$ cm; body weight: $59.76 \pm 1.92$ kg), and each group included five defenders, five midfielders, and four attackers. While the RT group did only routine soccer training, the HIPT group performed high-intensity plyometric training three days a week in addition to routine soccer training. In the study, body weight, stature, sprint (10 m, 30 m, and 40 m), jump (standing long jump, right and left-limb triple hop) and 5-0-5 change of direction speed test measurements of the soccer players were taken. Skewness and Kurtosis values obtained from the pre-test and post-test measurements were calculated to test whether the normality assumption of the study was met. A mixed measure two-way ANOVA test was used to determine the difference between the pre-test and post-test. The significance was set as $p < 0.05$. The results indicated that 8-week high-intensity plyometric training provided more improvement in measured performance parameters than routine soccer training. As a result, when the data obtained is examined, it can be said that HIPT had better values in sprint, jump, and change of direction parameters, so plyometric training was more beneficial for anaerobic parameters than routine soccer training in children. It is recommended that the trainers make their plans considering these results.

## INTRODUCTION

Due to the tactical changes and the success-oriented structure in today's soccer, it is necessary for the players to develop various characteristics (*Padrón-Cabo et al., 2021*). Today, most of the motor skills, depending on basic features and characteristics, are more prominent in soccer (*Clark et al., 2019*). While change of direction, anaerobic power, and ability to maintain that power at high-speed gain importance in soccer, endurance is needed even more (*Rivilla-García et al., 2019*). The systems of this game, which requires a high tackling power, force the players to frequently use their technical skills and aerobic-anaerobic power. Success in soccer demands superior physical, motor, and physiological performance levels as well as skill and ability (*Buchheit et al., 2013*).

In movements (walking, running, sprinting, jumping, and endurance) in soccer, many motoric performance characteristics must be developed due to the need for speed and frequent direction changes and the nature of the movements in the field (*Söhnlein, Müller & Stöggl, 2014*). High intensity running and repetitive sprint activities are considered to be very important indicators for successful performance (*Clemente et al., 2022*). The most important parameters for a soccer player are to be able to run faster than the opponent with or without the ball during the match and to move faster than the opponent while tackling to jump higher (*Ramírez-Campillo et al., 2014*). It is known that soccer players have aerobic base as they cover an average of 8.6–14.2 km during a match (*Dardouri et al., 2014*). However, considering that each player performs 1,000–1,400 short-term movements that last 4–6 s and each one performs sprints that end in an average of 2–4 s every 90 s, anaerobic performance is believed to be extremely important (*Köklü, Özkan & ve Ersöz, 2009*).

Anaerobic performance is mainly achieved through explosive type training that achieves results in the least amount of time. For this reason, the energy source that needs to be completed urgently is obtained from creatine phosphate (CP), adenosine triphosphate (ATP) and anaerobic glycolysis (*Asadi et al., 2016*). In anaerobic-based games such as football, boosts made by more than one method are obtained from the anaerobic energy system (*Zamparo et al., 2014*). Providing functional performance development of players towards the formation of anaerobic power also reveals the purpose of strength training. One of the factors affecting anaerobic performance and strength is muscles. Explosive muscle contractions, especially those produced by the knee extensors, have a very important role in the anaerobic performance measures of football players (*Beato et al., 2018*). During a football match, muscle power and strength are critical physical factors for successful participation. Plyometric training are widely implemented as training methodology for enhancing functional sports performance (*Stewart, Turner & Miller, 2014*).

Plyometric training, which can be performed by soccer players to increase explosive power is one of the best the training practises for the development of anaerobic power in soccer, and it can be applied with different arrangements aiming to accelerate the muscle response in combining strength and speed and increase in the number and quality of high-intensity movements (*Meylan & Malatesta, 2009*). These trainings in soccer meet the explosive strength requirement enables the eccentric muscle contractions and increase in

tension of the muscle. This increase in intramuscular tension increase muscle strength and thus the speed of movement (*Bompa, 2013*).

Soccer clubs, coaches, and soccer players are employing specific training models for parameters such as speed, agility, and strength, which are among the physical components of soccer players (*Loturco et al., 2015*). Research show that plyometric training is one of the most effective specific training methods that can directly affect match performance and develop the strength/power (*William & Kirubakar, 2018*). Additionally, if the aim of this success-indexed game is to increase the speed for athletic performance, anaerobic performance, jump height, explosive strength and maximal strength in the match, it has been proven that plyometric training comes first (*Váczi et al., 2013*).

Strength, and explosive strenght training have been reported to result in various physiological adaptations that enhance athletic performance (*Ferrari et al., 2008*). *Edwin & Gordon (2000)* reported that 8 weeks of a plyometric training improved sprint times, strength and anaerobic power (*Ferrari et al., 2008*). On the other hand, in order to increase the physical performance of football players, it is necessary to determine the physiological profiles of football players. Explosive power, force etc. when the training is based on physiological foundations for this profile, it is possible to increase the performance of the player (*Herrero et al., 2006*). Accordingly, it can be seen that all these physical characteristics can only reach the desired level thanks to a well-programmed training program. Preseason training is important in order to properly regulate the intensity and intensity of the training during the preparation season and to monitor the players' physical and physiological parameters well (*Buchheit et al., 2013*).

Preseason training is designed to develop players' physical capacities and prepare them for the various demands of the competitive season. For example, semi-professional and amateur soccer players who increased their aerobic capacity across a 8–10-week pre-season period were less likely to be injured during the subsequent in-season period (*Eliakim et al., 2018*). Preseason team training sessions were counted for each team, covering the period from the first training session of the season until the first competitive match (in the national league and international league). Pre-season team training sessions include a large variety of training types, for instance, match play, running, and fitness training (*Colby et al., 2017*).

Considering the relationship between regular soccer training (RT) and high-intensity plyometric training (HIPT) in the study, research show that there are links between high- intensity plyometric training and speed-jump performance. There are also limited number of studies that investigate this relationship. The problem statement of our research is whether high-intensity plyometric training makes a difference compared to regular training in the selected parameters. Our hypothesis is that high-intensity training makes a difference compared to regular training. Therefore, the aim of the study was to examine the effect of high-intensity plyometric training on some parameters in elite male soccer players in the U17.

**Table 1  Descriptive characteristics of the participants.**

| Variables | | n | x̄ | sd |
|---|---|---|---|---|
| Age (year) | RT Group | 14 | 16.71 | 0.47 |
| | HIPT Group | 14 | 16.64 | 0.50 |
| Stature (cm) | RT Group | 14 | 163.46 | 3.31 |
| | HIPT Group | 14 | 165.60 | 3.03 |
| Body weight (kg) | RT Group | 14 | 61.04 | 1.59 |
| | HIPT Group | 14 | 59.76 | 1.92 |

**Notes.**
x̄, Mean; sd, Standard deviation.

# METHOD

## Participants

A total of 28 male soccer players playing in the Elite Academy (U17) league of the Youth Development League of a team in the 1st league of Turkey participated in the study voluntarily (training experience; playing soccer in the elite academy team for at least 4 years). Each player spent ∼7 h in training plus one official match per week, and the team competed at the highest level for this age group. Each group consisted of five defenders, five midfielders and four attackers. The participation rate from the training of the RT and HIPT were 92.6% and 94.5% respectively.

In Table 1, it can be seen that the mean age of the soccer players in the RT group was $16.71 \pm 0.47$ years, the mean stature was $163.46 \pm 3.31$ cm, and the mean body weight was $61.04 \pm 1.59$ kg. Furthermore, the mean age of the HIPT group was $16.64 \pm 0.50$ years, the mean stature was $165.60 \pm 3.03$ cm, and the mean body weight was $59.76 \pm 1.92$ kg.

## Study design

In the study, randomized parallel matched-group design was adopted, and 28 volunteers were divided into two groups each including equal numbers: a regular training group (RT) and a high-intensity plyometric ıc training group (HIPT). The participants from the same team were randomly assigned to the groups. This study was conducted during the pre-season period of 2022–2023. It lasted ten weeks and consisted of a week of pre-testing, eight weeks of training interventions, and a week of post-testing (Table 2).

Data collection occured from June 2022–August 2022. For context, the study began after one weeks of pre-season commencement. The training intervention lasted eight weeks and we present the study's timeline in Fig. 1. Players were assessed 2 times over the period. Before each assessment, 48 h of rest were guaranteed regarding the last training session/match. The assessments were always performed on the same day of the week (Monday–Tuesday–Wednesday). The assessments started at 10 a.m.–12 p.m. The first assessment's average temperature and relative humidity were 27 °C and 2%, respectively. The second assessment's average temperature and relative humidity were 33 °C and 5%, respectively. Assessments occurred in full sun (Nobari et al., 2023).

As well as the regular training all the participants did, HIPT group also performed training interventions three days a week except for the first and last week. After completing

Peer*J*

**Table 2  Weekly training schedule.**

| Days | 1st week | 2nd week | 3rd week | 4th week | 5th week | 6th week | 7th week | 8th week | 9th week | 10th week |
|------|----------|----------|----------|----------|----------|----------|----------|----------|----------|-----------|
| **Monday** | Pre-test data collection | ST + HIPT | ST + HIPT | ST + HIPT | ST + HIPT | ST + HIPT | ST + HIPT | ST + HIPT | ST + HIPT | Post-test data collection |
| **Tuesday** | Pre-test data collection | ST | ST | ST | ST | ST | ST | ST | ST | Post-test data collection |
| **Wednesday** | Pre-test data collection | ST + HIPT | ST + HIPT | ST + HIPT | ST + HIPT | ST + HIPT | Cancelled training | ST + HIPT | ST + HIPT | Post-test data collection |
| **Thursday** | Day off | ST | ST | ST | ST | ST | ST | ST | ST | Resting day |
| **Friday** | Day off | ST + HIPT | ST + HIPT | ST + HIPT | ST + HIPT | ST + HIPT | ST + HIPT | ST + HIPT | ST + HIPT | Resting day |
| **Saturday** | Day off | Day off | Day off | Day off | Day off | Day off | Day off | Day off | Day off | Day off |
| **Sunday** | Day off | Day off | Day off | Day off | Day off | Day off | Day off | Day off | Day off | Friendly match |

**Notes.**

HIPT, High intensity plyometric training; ST, Soccer training.

**Table 3** Mean and standard deviation values of the measurement scores of the athletes participating in the study according to groups.

| Variables | Groups | Pre-Test | | Post test | | | Improvement rate % |
|---|---|---|---|---|---|---|---|
| | | $\bar{x}$ | sd | $\bar{x}$ | sd | $p$ | |
| Sprint 10 m (sec) | RT | 1.87 | 0.041 | 1.77 | 0.049 | 0.763 | 66 |
| | HIPT | 1.79 | 0.031 | 1.69 | 0.031 | | |
| Sprint 30 m (sec) | RT | 4.77 | 0.250 | 4.31 | 0.292 | 0.206 | 108 |
| | HIPT | 4.47 | 0.246 | 4.13 | 0.192 | | |
| Sprint 40 m (sec) | RT | 5.89 | 0.147 | 5.36 | 0.178 | 0.000* | 95 |
| | HIPT | 5.40 | 0.156 | 5.13 | 0.099 | | |
| Long jump (cm) | RT | 2.30 | 0.167 | 2.36 | 0.190 | 0.000* | 315 |
| | HIPT | 2.13 | 0.077 | 2.46 | 0.065 | | |
| Triple hop right (m) | RT | 7.12 | 0.057 | 7.33 | 0.112 | 0.166 | 14 |
| | HIPT | 6.84 | 0.204 | 7.14 | 0.199 | | |
| Triple hop left (m) | RT | 7.08 | 0.084 | 7.22 | 0.066 | 0.014 | 24 |
| | HIPT | 6.79 | 0.117 | 7.05 | 0.118 | | |
| 505 COD test (sec) | RT | 5.15 | 0.119 | 4.85 | 0.000 | 0.438 | 65 |
| | HIPT | 4.98 | 0.087 | 4.71 | 0.125 | | |

Notes.
*$p < .05$.

the interventions, they did regular soccer training during the 8 week period. The regular training lasted 90 min, and the training intervention session lasted approximately 20–25 min. All participants and their parents were informed about the procedures, requirements, benefits, and risks of the study beforehand and written consent was obtained before the study started. Moreover, this study was approved (approval code: E-37077861-200-38769) by the Iğdır University Ethics Committee (2021/21) and was conducted under the ethical guidelines of the Declaration of Helsinki for studying humans. In this study, we have followed all the Helsinki guidelines at all stages for human studies.

## Detailed schedule of training interventions
### Context of training intervention
The preseason consisted of twenty four sessions per week for 8 weeks. Microcycles of weekly training are presented in Table 2. In the preseason, most training sessions included (in this order): submaximal aerobic training (65–75% HRmax), interval runs (80–95% HRmax), high/ medium interval training. Also, the intensity of exercise was selected in the range between 70 to 95%) and strength training. The strength training performed in the pre-season was circuit based and was the same for all players. In addition to regular training, players participated in twenty four HIPT during the preseason. During the season,

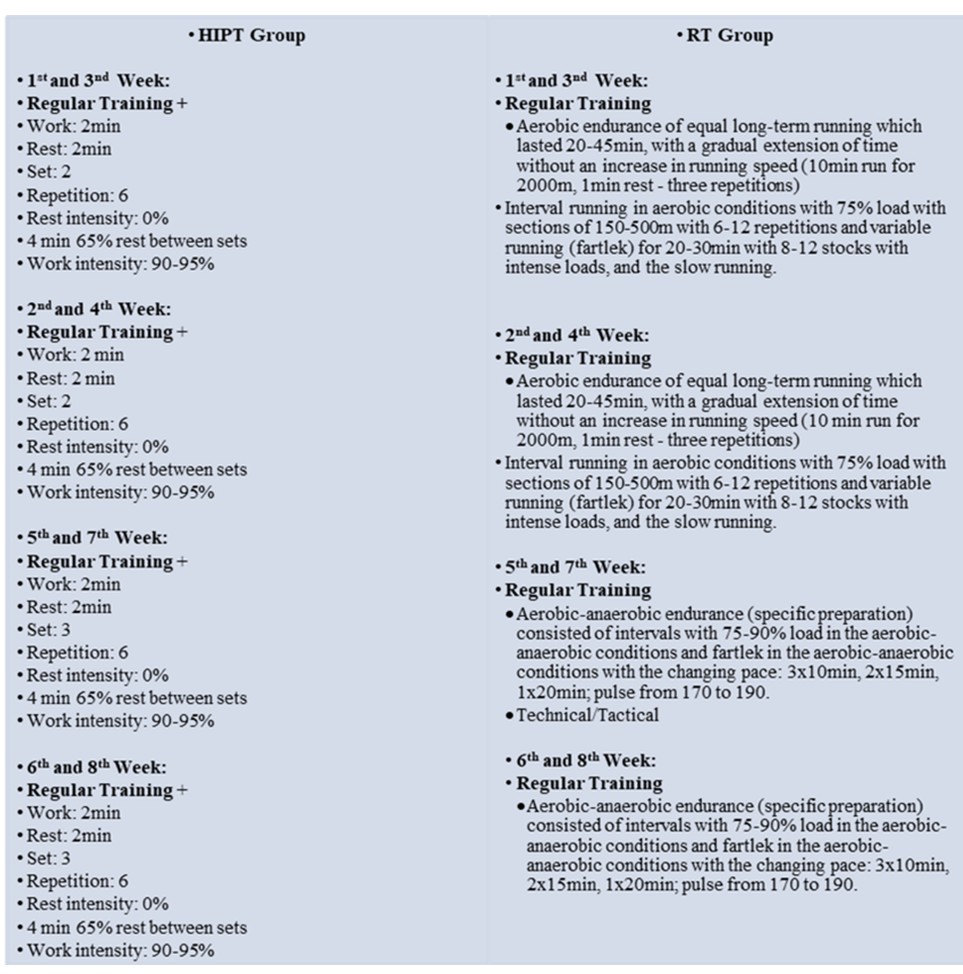

**Figure 1  A detailed weekly training schedule.**

participants trained five times per week, along with tactical one match. Table 2 and Fig. 1 shows the details of each training session during the in pre-season period and the scheduling of the research protocol within the training schedule.

   Consequently, all training sessions were held on natural grass with stock shoes and soccer training uniforms. The pattern of formation of the names of movements is by the study of *Loturco et al. (2020)*, but in this study, the intensity, rest time, the number of repetitions and duration of training intervention vary. Movement intensity, volume and speed were considered the same during the research period.

   Figure 1 shows the detailed schedule of the training interventions which were performed in a 8 weeks period, the participants had five training sessions each week, and in three of these sessions they also performed the training interventions. These sessions were on Mondays, Wednesdays, and Fridays. First, the participants started the training with a standard warm up targeting (Fig. 2). The HIPT group performed a change of direction and sprint training (three sets of three short shuttle runs with four change of direction each, for an amount of thirty change of direction) and 4 × 5 drop jumps over 60 cm height followed

| |
|---|
| 5 minute-jogging with 40-50% of the max. heart rate |
| 5 frontal-lateral hopping: One Step / Rabbit Hops / Two-Foot Side /Hops-Down The Side / Side Shuffle |
| 5 mobilization: In-line lunge / Shoulder Mobility / Active Stright Leg Rise / Trunk Stability Push Up / Rotary Stability |
| 5 upper extremity static stretching exercises:<br>*Stretching the quadriceps by pulling the heel of the foot towards the hip<br>*With the body in an upright position, bending the knees, holding the foot with one hand and pulling it towards the abdomen<br>*Body closes towards the ankles without bending the knees in the upright position and waiting by holding the ankles<br>*In long sitting on the floor, legs spread and stretching forward<br>* In the long sitting position, the upper part of the body (trunk) is stretched by leaning forwards and touching the fingertips |

**Figure 2  Spesific warm up protocol.**

by a subsequent jump over two obstacles (15 cm height), 4 × 6 horizontal jumps, as well as 4 × 6 jumps over 15 cm obstacles.

## Specific warm up protocol

The players underwent a 20-min warm up session led by a licensed strength and conditioning coach involving stretching, the classical warm-up procedure applied in the study includes 5 minute-jogging with 40–50% of the maximum heart rate, followed by five frontal-lateral hopping, five mobilization, five upper extremity static stretching exercises and familiarization with the test, respectively (Fig. 2). The movements were applied for two sets and 15 s.

In addition, athletes did a standard warm-up focusing on lower limbs which consisted of self- paced low-intensity running, lower-limb dynamic stretching, and reactive strength exercises before the tests.

## Data collection methods
### Body weight

Body weight measurements of the participants were obtained in kg using the Direct Segmental Multi-frequency–Bioelectrical Impedance Analysis Method (DSM-BIA) (Inbody 270; Biospace, Cerritos, CA, USA), with the electronic scale integrated into the Bioelectrical impedance analyzer, while they were barefoot and in anatomical posture wearing only shorts and a T-shirt (*Lukaski, 2003*).

### Stature

Body height of the participants was obtained in cm with a wall-mounted stadiometer (Holtain Ltd, Crymych, UK) while they were in anatomical posture, bare feet, heels together, holding their breath, head in the frontal plane, with the headboard touching the vertex point.

### Sprint tests

The *10 m-30 m-40 m* straight sprint tests were performed by the participants, and the performance was measured using the Fusion SmartSpeed Timing Gates System (Fusion

 

Sport, Coopers Plains, QLD, Australia) connected to a wireless computer. The system is used to measure the sprint time and consists of gates, each of which has a photocell with an infrared transmitter and a light reflector, and a radio frequency identification reader to identify the athlete.

There were eight gates in 40-meter run over a straight-line test. The distance between the photocell and the light reflector was 2 m, and the gates were placed at 10 m, 30 m, and 40 m away from the start line, and the 40 m gate indicated the finish line. When the athletes crossed these gates, with the help of the IR beam their times were recorded. The athletes started from a standing position after the signal was given (*Sporiš et al., 2011*).

### Standing long jump test
In the standing long jump (SLJ) test, athletes jumped horizontally from a standing still position and tried to move their body forward as far as possible. The total jump distance from the take-off line to the mark made on landing by the heel of the athlete was recorded in cm with a tape measure. The atheltes were not allowed to take steps backward or perform preparatory hops/runs (*Sgrò et al., 2017*).

### Triple hop test
Three triple hops were performed by the participants on artificial turf, and they did the test for each leg and were allowed to swing their arms during. The mean of each test score for each leg was calculated and recorded for data analysis. The sideline of a soccer pitch used for the test, and each yard of a testing area was outlined by paint. The athletes were instructed to put their foot behind the start line and hop forwards three times as far and quickly as possible to decrease contact time with ground (GCT). On the third and the final hop, they were asked to stay in that position for 3 s. When they could not 'stick', the attempt was declared void, and the athlete was asked to repeat the jumps after a 90 s rest. There was a 90 s break between each trial, and the trials were performed alternating the leg (*i.e.,* 1st trial = left leg, 2nd trial = right leg, 3rd trial = left leg and so on) (*Lloyd et al., 2020*).

### 505 Change-of-direction speed test
The methodology used for the 505 change-of-direction speed test (Fig. 3) was used as per established methods, with one timing gate (Fusion Sports, Coopers Plain, QLD, Australia) positioned to record time. The modified 5-0-5 test protocol was employed to measure the players' COD time and COD deficit. The test consists of starting in a standing position (foot split) and accelerating over a 10-m distance before performing two COD of 180° (from A to C point was 5+5 m). The time from the final 10-m (5+5 m) is recorded using two pairs of photocells (Fusion Sports, Coopers Plain, QLD, Australia). The photocell height was adjusted based on the height of the player's hip. The players were allowed to use the preferred leg for braking and turning movements. However, they were always asked to use the same leg. The same instruction was used for the foot in front at the starting position. Each participant performed two trials, with a rest period of three minutes. The COD time (s) was obtained for each trial. The smallest time was used for further statistical procedures (*Nobari et al., 2020*; *Nimphius et al., 2016*).

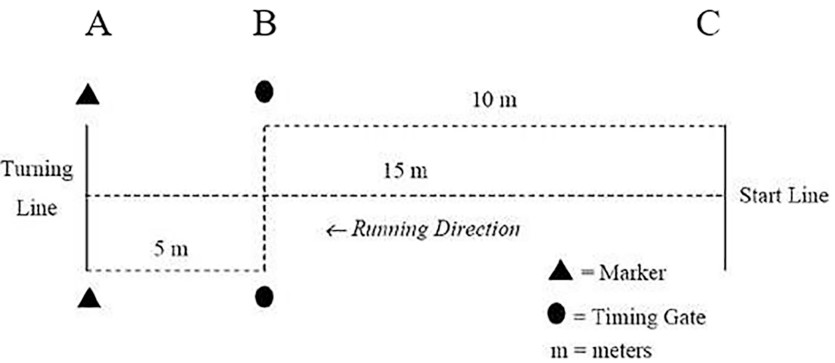

**Figure 3 Change of direction speed test.**

| Days | Measurements |
|---|---|
| • Sunday | Day Off |
| Monday | (10:00) Height Measurement and Body Composition Analyse |
| Tuesday | (16:00) Standing Long Jump (SLJ) and Triple Hop Test (THT) |
| Wednesday | (10:00) Sprint and 505 Change of Direction Tests (COD) |
| Thursday | Day Off |
| Friday | Day Off |

**Figure 4 Test schedule.**

### *Test schedule*

Pre-test and post-test measurements were completed on the first and the last week from Monday to Wednesday. In the pre-test in pre-season, age, stature, body weight measurements were taken in the morning at 10:00. On the second day at 10:00 in the morning, long jump test and triple hop test were conducted for right and left limbs separately. On the third day at 10:00 am, the athletes performed sprint and change of direction (COD) tests. The same procedure was also completed on the last week. Before the tests started, the participants were asked to not to change their usual dietary intake on the assessment days, and they were informed about the test procedures and divided into two groups (Fig. 4).

### Research model

Quasi-experimental design was used in the study. Since the process of determining unbiased sampling is generally difficult in sports and health fields, the quasi-experimental design, which is considered within experimental designs, is more preferred (*McMillan*

**Table 4  ANOVA results of 10 m sprint (sec) performance scores of the athletes participating in the study.**

| Source of variance | ss | Sd | ms | ƒ | p |
|---|---|---|---|---|---|
| Between groups | 0.154 | 27 | | | |
| Group | 0.091 | 1 | 0.091 | 37.72 | 0.000[*] |
| Error | 0.063 | 26 | 0.002 | | |
| Within groups | 6.592 | 28 | | | |
| Measurement (PreTest-PostTest) | 0.156 | 1 | 0.156 | 226.26 | 0.000[*] |
| Measurement*Group | 6.42 | 1 | 6.429 | 0.093 | 0.763 |
| Error | 0.018 | 26 | 0.001 | | |

Notes.
   ss, Sum of squares; sd, Standart deviation; ms, Mean of squares; f, Variance value.
   *$p < .05$.

& Schumacher, 2010). The main difference of the quasi-experimental design from the full experimental design is that the sample is not randomly determined in the quasi-experimental design (Çoban, 2017).

## Data analysis

Whether the existing data are normally distributed was determined by applying skewness and kurtosis tests. Skewness and kurtosis values between +3 and −3 are generally accepted for normal distribution (Kalaycı, 2009). Since the skewness (+1,31 to −1,29) and kurtosis (+2,35 to −1,32) values of all analyzed measurements were within the above ranges, It was accepted to be normally distributed. For the purposes of the study, the Mixed Measure Two-Way ANOVA Test was used to determine the difference between the pre-test and post-test. In the statistical analysis of the data obtained in the study, the margin of error was taken as $p < .05$. For the purposes of the study, the mixed measure two-way ANOVA test was used to determine the difference between the pre-test and post-test. In the statistical analysis of the data obtained in the study, the margin of error was taken as $p < .05$.

## RESULTS

In this section, after giving the descriptive information (Table 3) of the athletes participating in our study, statistical results regarding the parameters 10 m sprint (sec) (Table 4), 30 m sprint (sec) (Table 5), 40 m sprint (sec) (Table 6), long jump (cm) (Table 7), triplehop right (m) (Table 8), triplehop left (m) (Table 9), 505 cod test (sec) (Table 10) scores are given.

It can be seen in Table 3 that there is a statistically significant difference between the groups in the 40 m sprint (sec) and long jump (cm) parameters ($p < 0.05$). There is no statistically significant difference in other parameters ($p > 0.05$). When looking at the improvement rates, there is a significant improvement (15%) in the HIPT group compared to the RT group in the long jump parameter. There is also a significant improvement in the 40m sprint parameter in the RT group compared to the HIPT group (5%, 9% respectively).

In Table 4, a significant difference is found between the athletes in the RT (football training) and HIPT (high-intensity plyometric training) groups in terms of their 10-metre

**Table 5** ANOVA results of sprint 30 m sprint (sec) performance scores of the athletes participating in the study.

| Source of variance | ss | sd | ms | f | p |
|---|---|---|---|---|---|
| Between groups | 3.1 | 27 | | | |
| Group | 0.780 | 1 | 0.780 | 8.74 | 0.007[*] |
| Error | 2.32 | 26 | 0.089 | | |
| Within groups | 3.21 | 28 | | | |
| Measurement (PreTest-PostTest) | 2.26 | 1 | 2.26 | 66.60 | 0.000[*] |
| Measurement*Group | 0.057 | 1 | 0.057 | 1.68 | 0.206 |
| Error | 0.885 | 26 | 0.34 | | |

Notes.
ss, Sum of squares; sd, Standart deviation; ms, Mean of squares; f, Variance value.
[*]$p < .05$.

**Table 6** ANOVA results of 40 m sprint (sec) performance scores of the athletes participating in the study.

| Source of variance | ss | sd | ms | f | p |
|---|---|---|---|---|---|
| Between groups | 2.68 | 27 | | | |
| Group | 1.86 | 1 | 1.86 | 59.12 | 0.000[*] |
| Error | 0.819 | 26 | 0.031 | | |
| Within groups | 2.76 | 28 | | | |
| Measurement (PreTest-PostTest) | 2.20 | 1 | 2.20 | 176.28 | 0.000[*] |
| Measurement*Group | 0.233 | 1 | 0.233 | 18.61 | 0.000[*] |
| Error | 0.325 | 26 | 0.013 | | |

Notes.
ss, Sum of squares; sd, Standart deviation; ms, Mean of squares; f, Variance value.
[*]$p < .05$.

**Table 7** ANOVA results of long jump (cm) performance scores of the athletes participating in the study.

| Source of variance | ss | sd | ms | f | p |
|---|---|---|---|---|---|
| Between groups | 0.912 | 27 | | | |
| Group | 0.018 | 1 | 0.018 | 0.520 | 0.477 |
| Error | 0.894 | 26 | 0.034 | | |
| Within groups | 0.883 | 28 | | | |
| Measurement (PreTest-PostTest) | 0.564 | 1 | 0.564 | 195.97 | 0.000[*] |
| Measurement*Group | 0.244 | 1 | 0.244 | 84.94 | 0.000[*] |
| Error | 0.075 | 26 | 0.003 | | |

Notes.
ss, Sum of squares; sd, Standart deviation; ms, Mean of squares; f, Variance value.
[*]$p < .05$.

**Table 8 ANOVA results of triplehop right (m) performance scores of the athletes participating in the study.**

| Source of variance | ss | sd | ms | f | p |
|---|---|---|---|---|---|
| Between groups | 1.70 | 27 | | | |
| Group | 0.785 | 1 | 0.785 | 22.12 | 0.000[*] |
| Error | 0.923 | 26 | 0.035 | | |
| Within groups | 1.31 | 28 | | | |
| Measurement (PreTest-PostTest) | 0.939 | 1 | 0.939 | 70.67 | 0.000[*] |
| Measurement*Group | 0.027 | 1 | 0.027 | 2.03 | 0.166 |
| Error | 0.345 | 26 | 0.013 | | |

**Notes.**

ss, Sum of squares; sd, Standart deviation; ms, Mean of squares; f, Variance value.

[*]$p < .05$.

**Table 9 ANOVA results of triplehop left (m) performance scores of the athletes participating in the study.**

| Source of variance | ss | sd | ms | f | p |
|---|---|---|---|---|---|
| Between groups | 1.07 | 27 | | | |
| Group | 0.734 | 1 | 0.734 | 56.64 | |
| Error | 0.337 | 26 | 0.013 | | |
| Within groups | 0.751 | 28 | | | |
| Measurement (PreTest-PostTest) | 0.530 | 1 | 0.530 | 78.82 | 0.000[*] |
| Measurement*Group | 0.046 | 1 | 0.046 | 6.87 | 0.014 |
| Error | 0.175 | 26 | 0.007 | | |

**Notes.**

ss, Sum of squares; sd, Standart deviation; ms, Mean of squares; f, Variance value.

[*]$p < .05$.

**Table 10 ANOVA results of 505 cod test (sec) performance scores of the athletes participating in the study.**

| Source of variance | ss | sd | ms | f | p |
|---|---|---|---|---|---|
| Between groups | 0.663 | 27 | | | |
| Group | 0.323 | 1 | 0.323 | 24.66 | |
| Error | 0.340 | 26 | 0.013 | | |
| Within groups | 0.751 | 28 | | | |
| Measurement (PreTest-PostTest) | 1.28 | 1 | 1.12 | 193.53 | 0.000[*] |
| Measurement*Group | 0.004 | 1 | 0.004 | 0.620 | 0.438 |
| Error | 0.152 | 26 | 0.006 | | |

**Notes.**

ss, Sum of squares; sd, Standart deviation; ms, Mean of squares; f, Variance value.

[*]$p < .05$.

running performance scores [F(1. 26) = 37.72, $p < .05$]. It was determined that the difference between the pre-test and post-test 10-metre running performance scores of the athletes included in the study was significant [F(1. 26) = 226.26, $p < .05$]. As seen in the table, it was determined that the measurement*group joint effect of the test results of the athletes' 10-metre running performance scores was not significant [F(1. 26) = .093, $p > .05$].

A significant difference is found in Table 5 between the athletes in the RT and HIPT groups in terms of the score they received from the 30 m sprint (sec) performance [F(1. 26) = 8.74, p<.05]. It was determined that the difference between the pre-test and post-test 30 m sprint (sec) run performance scores of the athletes included in the study was significant [F(1. 26) = 66.60, p<.05]. This result shows that the sprint 30 m sprint (sec) performance scores of the athletes increased in the process. As seen in the table, it was determined that the measurement*group joint effect of the test results of the athletes' 30 m sprint (sec).

A significant difference is found in Table 6 between the athletes in the RT and HIPT groups in terms of their 40 m sprint (sec) performance scores [F(1. 26) = 59.12, $p < .05$]. It was determined that the difference between the pre-test and post-test 40 m sprint (sec) performance scores of the athletes included in the study was significant [F(1. 26) = 176.28, $p < .05$]. This result shows that the 40 m sprint (sec) performance scores of the athletes increased in the process. As seen in the table, it was determined that the measurement*group joint effect of the test results of the athletes' forty metre run performance scores was significant [F(1. 26) = 18.61, $p < .05$]. Therefore, it is seen that the training applied to the athletes has an effect on the 40 m sprint (sec) performance scores of the athletes.

In Table 7, no significant difference is found between the athletes in the RT and HIPT groups in terms of their long jump (cm) performance scores [F(1. 26) = .520, $p > .05$]. It was determined that the difference between the pre-test and post-test long jump (cm) performance scores of the athletes included in the study was significant [F(1. 26) = 195.97, $p < .05$]. This result shows that the long jump (cm) performance scores of the athletes increased in the process. As seen in the table, it was determined that the measurement*group joint effect of the test results of the long jump (cm) performance scores of the athletes was significant [F(1. 26) = 84.94, $p < .05$]. Therefore, it is seen that the training applied to the athletes has an effect on the long jump (cm) performance scores of the athletes.

Table 8 shows a significant difference was found between the athletes in the RT and HIPT groups in terms of their triplehop right (m) performance scores [F(1. 26) = 22.12, $p < .05$]. It was determined that the difference between the pre-test and post-test triplehop right (m) metre running performance scores of the athletes included in the study was significant [F(1. 26) = 70.67, $p < .05$]. This result shows that the triplehop right (m) performance scores of the athletes increased in the process. As seen in the table, it was determined that the measurement*group joint effect of the test results of the triplehop right (m) running performance scores of the athletes was not significant [F(1. 26) = 2.03, $p > .05$]. Therefore, it is seen that the training applied to the athletes had no effect on the triplehop right (m) performance scores of the athletes.

In Table 9, a significant difference was found between the athletes in the RT and HIPT groups in terms of their triplehop left (m) performance scores [$F_{(1, 26)} = 56.64$, $p < .05$]. It was determined that the difference between the pre-test and post-test triplehop left (m) metre running performance scores of the athletes included in the study was significant [$F_{(1, 26)} = 78.82$, $p < .05$]. This result shows that the triplehop left (m) performance scores of the athletes increased in the process. As seen in the table, it was determined that the measurement*group joint effect of the test results of the triplehop left (m) running performance scores of the athletes was not significant [$F_{(1, 26)} = 6.87$, $p > .05$]. Therefore, it is seen that the training applied to the athletes had no effect on the triplehop left (m) performance scores of the athletes.

In Table 10, a significant difference was found between the athletes in the RT and HIPT groups in terms of the score they received from the 505 COD test (sec) performance [$F_{(1, 26)} = 24.66$, $p<.05$]. It was determined that the difference between the pre-test and post-test 505 COD test (sec) running performance scores of the athletes included in the study was significant [$F_{(1, 26)} = 193.53$, $p<.05$]. This result shows that the athletes' 505 COD test (sec) performance scores increased in the process. As seen in the table, it was determined that the measurement*group joint effect of the test results of the athletes' 505 COD test (sec) performance scores was not significant [$F_{(1, 26)} = .620$, $p > .05$]. Therefore, it is seen that the training applied to the athletes has no effect on the 505 COD test (sec) performance scores of the athletes.

# DISCUSSION

It is known that athletic performance measurements and different training models are important factors for youth soccer players in elite level who are in their development stages to achieve success. This study was conducted to reveal the effect of 8-week high-intensity plyometric training on sprint, jump and change of direction parameters in young male (U17 Elite A-league) soccer players. The results indicated that this training provided more improvement in measured performance parameters than routine soccer training training (Tables 3–10). This shows that high-intensity plyometric training can be included in the necessary training units as a specific training unit, starting from the lower age groups, taking into account the developmental stages of children. To collect data for the study, pre-test and post-test design was used to measure the parameters, and the results were discussed taking the literature into account.

## Sprint performance

Sprinting ability is accepted as the main determining factor in sportive performance in soccer. When we analyze soccer matches, the ability of continually sprint and do so by changing direction is considered as a determinant of high-level performance for coaches and researchers, especially in team sports such as soccer. Moreover, this is also accepted as an indicator of the fitness level of the athletes (*Wong, Chan & Smith, 2012a*).

According to the results obtained from the study, when the pre-test and post-test values between the high-intensity plyometric training group and the regular training group were examined, it was determined that there was a significant difference between all sprint values

except for the 30 m sprint value of the post-test. When the mean values are compared, it is shown that the post-test values of the high-intensity plyometric training group were better. In addition, similar improvement rate was observed in the 10m sprint performance (6%; 6%), while RT showed greater improvement in the 30 m (10%; 8%) and 40 m (9%; 5%) sprint performances (Tables 3–6).

*Boraczyński & Urniaz (2008)* examined the effect of plyometric training for 8 weeks on strength and speed parameters and observed a positive increase in the leg strength and speed characteristics of the athletes. *Fernandez-Fernandez et al. (2016)* reported that 8-week plyometric training with young tennis players had a positive and significant improvement in the speed characteristics. *Loturco et al. (2015)* stated that there was an improvement and statistical difference between the 30 m pre- and post-test values of 15 athletes who participated only in plyometric training for 8 weeks. In a study in which mixed method adopted, it was seen that plyometric exercises combined with maximal strength and high-intensity resistance training improved the sprint performance of athletes (*Villarreal, Requena & Newton, 2010*). Furthermore, *Meylan & Malatesta (2009)* concluded that 6-week plyometric exercises improve 30 m sprint performance of soccer players. *Asadi et al. (2018)* also stated that plyometric training should be applied together with other training programs in order to improve sprint speed. *Wong et al. (2012b)* stated that plyometric training combined with weight training significantly increased the speed of athletes. In their study *Diallo et al. (2001)* reported that sprint performance significantly increased in the training group using a sprint bike compared to the control group. As can be seen from the above study results, which are similar to our study findings, 6–8 weeks of plyometric training has positive effects on the sprint parameter.

The results showed that the increase in the values of the high-intensity training group affects the improvement of the speed characteristics of plyometric soccer training. Considering that the speed characteristics is the least improvable feature among the biomotor abilities that can be improved, it can be accepted that developing strength with the help of plyometric training makes a positive contribution to the existing speed potential. Furthermore, it can be said that plyometric training can improve short-distance sprint speed.

### Long jump performance

The standing long jump is a test that requires the involvement of multiple joints and is widely used to evaluate the explosive power of the leg muscles (*Moresi et al., 2011*). In this study, explosive leg strength of young soccer players was evaluated with dominant, non-dominant, and two-footed standing long jump tests.

The results of this study present that while there is a statistically significant difference in the pre-test values between the groups, there is no difference in the post-test values (Table 7). When the mean values are compared, it is seen that the HIPT post-test measurement values were better. In addition, HIPT showed much more improvement than RT (3% and 15% respectively) (Table 5). When the literature related to this parameter is examined, in a study in which 6-week mixed training program was applied to the soccer players during in-season period, no significant difference was found in terms of the standing long jump

values (*Chtara et al., 2017*). There are studies with similar results regarding the results we obtained, and in a study comparing individual athletes and team, no significant difference was found in standing long jump values (*Koç & Aslan, 2010*). *Wang & Zhang (2016)* obtained similar results to our research in their study on plyometric training in elite soccer players.

Moreover, it is thought that high-intensity plyometric training led to an increase in jump distance in soccer players in this study. Many studies have used at least one jump test to determine the effects of plyometric training on athletic performance. For example, in the study conducted by *Buchheit et al. (2010)*, a statistically significant difference was observed in counter movement jumping (CMJ) in the plyometric training group compared to the sprint training group. Similarly, *Meylan & Malatesta (2009)* found that there was a statistically significant improvement in the CMJ training group compared to the control group. *Michailidis et al. (2013)* stated that when compared to the control group, there was a significant improvement in tests such as SJ, CMJ, and depth jump (DJ) in the training group, and *Rubley et al. (2011)* reported that the training group improved significantly in vertical jump (VJ) performance compared to the control group. *Thomas, French & Hayes (2009)* examined the effects of both DJ and CMJ plyometrics training on strength and agility in young soccer players and reported that both training groups showed significant improvement in CMJ.

*Chelly et al. (2010)* determined that an 8-week plyometric training period improved the standing long jump distance in the experimental group consisting of active healthy individuals compared to the control group. Furthermore, it was stated that compared to the control group, a 10-week plyometric training had a 2.8% increase in standing long jump distance performance in the experimental group, which included active healthy individuals (*Markovic et al., 2007*). In our study, when the mean values are compared, it is shown that the post-test values were better and showed an increase, and the results obtained correlated with the literature. According to the results obtained from this research, it was determined that high- intensity soccer training improved the standing long jump performance compared to routine training. In addition, the fact that both research groups were in the developmental age may have ensured that there was no significant difference in the post-test values between the groups.

## Triple hop (right and left) performance

The triple jump test requires a combination of muscle strength, power, and balance (*Noyes, Barber & Mangine, 1991*). This test reflects the lower extremity muscle strength and power of soccer players (*Hamilton et al., 2008*).

According to the results obtained from the study, there is a statistically significant difference between the groups in term of the pre-test and the post-test values (Tables 8 and 9). When the mean values are compared, it is shown that the RT post-test values were better. However, HIPT showed much more improvement than RT (right: 4% and 1%; left: 4% and 2%, respectively) (Table 5). *Ozbar, Ates & Agopyan (2014)* found that in female athletes in the plyometric training (PT) group, dominant leg increased by 12.1%, the non-dominant leg increased by 15.7%, and in the control group, the dominant leg increased by 4.3% and

the non-dominant leg increased by 6.6% in terms of the triple hop for distance (THD) performance. This situation reveals the effect of PT from different perspectives. It was observed that PT made a greater difference in the jump distance of the non-dominant leg compared to the dominant leg, especially in the THD test. It is reported that this may be related to the significant increase in performance in all jump parameters and the common technical and conditional exercises performed on other days in the control group (*Ozbar, Ates & Agopyan, 2014*). In this study, which is similar to our results, the higher percentage of development in this study may be due to gender. Furthermore, inter-group analysis after a 12- week plyometric training of professional female soccer players by *Nonnato et al. (2022)* reported a positive change in terms of the triple jump test results of dominant limb ($p = 0.031$; medium effect size) and non-dominant limb ($p = 0.021$; medium effect size). *Beato et al. (2018)* reported that there were improvements between the groups in the triple jump test results after 6 weeks of plyometric training, but there was no significant correlation. According to the results obtained in this study, it is shown that there was no correlation with the literature in terms of this parameter. In addition, it is seen that further studies are needed to be conducted to eliminate these contradictions and to determine the effects of plyometric training on triple jump performance.

## 505 COD performance

Agility, which is defined as a rapid change in the speed and direction of movement in response to an external stimulus (*Tabacchi et al., 2018*), is a concept based athletic performance and the change in decision-perception and frontal speed in soccer and it is affected by these factors when the quality of a player is evaluated (*Sheppard & Young, 2006*). Change-of-direction time (Fig. 3), namely the 5-0-5 test time, is nearly perfectly correlated with linear acceleration in small distances such as 5-, 10-m, and 20-m (*Nobari et al., 2023*).

According to the results obtained from this study, there was a statistically significant difference between the groups in terms of the pre-test and the post-test values (Table 10). When the mean values are compared, it is seen that the HIPT post-test measurement values were better. In addition, HIPT and RT showed similar improvement rates (6% and 5%, respectively) (Table 3).

Similar to our study results, in a study conducted with different age groups, it was stated that there was a significant difference in favor of the experimental group in the mean values of agility in the pre- and post-tests (*Atacan, 2010*). In another study conducted with 15–16 year-old soccer players, it was determined that the post-test mean values of agility were better in favor of the experimental group (*Hazır, Mahir & Açıkada, 2010*). In addition, in a study similar to the results of our research, they found that after particularly arranged plyometric training, agility post-test values increased in favor of the experimental group compared to the control group (*Pienaar & Coetzee, 2013*). In a study conducted with soccer players playing in development leagues, it was determined that there was a significant increase in the agility values of male soccer players who participated in plyometric training during the preparation period (*Padrón-Cabo et al., 2021*). In their study, *William & Kirubakar (2018)* examined the relationship between the plyometric and agility values of soccer players also obtained similar results with our study.

Taking these results into consideration, it is thought that the increase in agility performance of the high-intensity plyometric training group compared to the regular training group is due to the application of plyometric training in different combinations together with soccer training. However, it can be said that the positive improvement in the regular training group and the soccer training program have a positive effect on the selected parameter.

The present study contains limitations that need to be taken into account. First, the food consumption records and hydration levels of the participants were not monitored throughout the 10-week study period. To address this limitation, a 24-hour retrospective food record could have been employed to track the participants' food consumption for two weekdays and one day on the weekend, spanning a total of three days. This would have allowed for the calculation of their energy intake. Another limitation of our study is that it was exclusively conducted with young male athlete participants. This narrow focus on young male athlete participants makes it challenging to generalize the results, particularly in relation to the adult male athlete population. Future research should consider including both male and female participants to obtain a more comprehensive understanding of the findings and their applicability to a broader population.

## CONCLUSIONS

Jumping and speed are directly linked with leg and hip strength, and various methods have been developed to improve particularly leg strength. The most commonly used one of these is the plyometric training (*Chaabene & Negra, 2017*). When plyometric training is performed regularly and correctly, it contributes to an increase in performance in sports branches (*i.e.,* soccer, handball, volleyball) where jump and speed parameters are very important (*Aksović et al., 2021*). In addition, it is thought that the studies examining the effect of plyometric training designed for lower and upper extremities using various techniques or on different floors/surfaces can provide new information to sports sciences (sand, land, water) on biomotor abilities in different age groups. In addition, the relationship between body composition and jump in young male soccer players is an area open to research.

As a result, when the data obtained is examined, it can be said that HIPT had better values in sprint, jump, and change of direction parameters, so plyometric training was more beneficial for anaerobic parameters than routine soccer training in children. It is recommended that the trainers make their plans considering these results.

## ACKNOWLEDGEMENTS

We would like to thank all the athletes and coaching staff for their support during the study measurements.

### Funding

The authors received no funding for this work.

### Competing Interests

The authors declare there are no competing interests.

### Author Contributions

- Mehmet Söyler conceived and designed the experiments, performed the experiments, analyzed the data, prepared figures and/or tables, authored or reviewed drafts of the article, and approved the final draft.
- Raif Zileli conceived and designed the experiments, performed the experiments, analyzed the data, prepared figures and/or tables, authored or reviewed drafts of the article, and approved the final draft.
- Yunus Emre Çingöz conceived and designed the experiments, performed the experiments, analyzed the data, prepared figures and/or tables, authored or reviewed drafts of the article, and approved the final draft.
- Gökmen Kılınçarslan conceived and designed the experiments, performed the experiments, analyzed the data, prepared figures and/or tables, authored or reviewed drafts of the article, and approved the final draft.
- İdris Kayantaş conceived and designed the experiments, performed the experiments, analyzed the data, prepared figures and/or tables, authored or reviewed drafts of the article, and approved the final draft.
- Tolga Altuğ conceived and designed the experiments, performed the experiments, analyzed the data, prepared figures and/or tables, authored or reviewed drafts of the article, and approved the final draft.
- Selim Asan conceived and designed the experiments, performed the experiments, analyzed the data, prepared figures and/or tables, authored or reviewed drafts of the article, and approved the final draft.
- Musa Şahin conceived and designed the experiments, performed the experiments, analyzed the data, prepared figures and/or tables, authored or reviewed drafts of the article, and approved the final draft.
- Alper Cenk Gürkan conceived and designed the experiments, authored or reviewed drafts of the article, and approved the final draft.

### Human Ethics

The following information was supplied relating to ethical approvals (i.e., approving body and any reference numbers):

In the meeting of Iğdır University Scientific Research and Publication Ethics Committee dated 07.07.2021 and numbered 2021/21, this study was carried out in accordance with the 10/1 of the Iğdır University Scientific Research and Publication Ethics Directive. It has been decided that it is in compliance with scientific research and publication ethics (Approval Code: E-37077861-200-38769).

## Data Availability

The raw data are available in the Supplemental Files.

## Supplemental Information

Supplemental information for this article can be found online at http://dx.doi.org/10.7717/peerj.16648#supplemental-information.

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
