# Peer review of "The effect of high-intensity plyometric training on anaerobic performance parameters: a pilot study in U17 elite A league"

_PeerJ, doi:10.7717/peerj.16648_

## Round 0.1 · original submission · Major Revisions

The submitted article requires some major revisions. Please consider the following comments and revise accordingly:

Line 98-111. Please include the participants’ sex in this section and avoid or reduce the repetitive parts (e.g., line 108-111).
Line 132: Identify is misspelled.
Line 161-171: The test described does not seem to be the 505, and in the citation there is no indication of the 505 test. Please carefully check the test name and provide a citation.
Line 179: Please consider moving the study design paragraph after the participant section.
Line 254-259: Please, carefully follow PeerJ guidelines regarding statistical analysis (e.g., correction for type one error inflation), which have to be met to make the article acceptable for publication. Additionally, I was wondering why the authors did not use a more comprehensive statistical analysis (e.g., mixed ANOVA or MANOVA), which is able to provide direct information on the interaction effects between the group and time. These approaches could allow the authors to directly compare the changes in the two groups and protect against type one error inflation. In order to facilitate the interpretation of the study results and make them ready to be included in meta-analyses and systematic reviews, the authors should provide the effect size for the statistical test performed and report the exact p value to the third decimal point.
Line 261: Consider changing it into results.
Line 268-270 and 279-281: These sentences seem to be interpreting the results a bit too much for the results section, making them more appropriate for the discussion section. Please consider rephrasing them.
Line 289-291: This sentence is a quite strong statement. Consider making it softer in the context of the results obtained in the present study.

·

Basic reporting

First, thank you for the opportunity to comment on the practical application of high-intensity plyometric training. The work describes in detail the research procedures and the differences in the results in the analyzed groups. For the work to be accepted and made public, I will suggest a few minimal changes to the position below to improve the quality.

Paragraph 82 -91- Suggests deleting this part of the text. The body composition results of the players have not appeared anywhere in the work, so we have no way to comment on this issue.
However, you can show this element as a deficit at work and suggest it as a further research direction.

Paragraphs 99 - 103. In the section describing the group of the examined players, there is no information about the training experience (e.g. years at the Football Academy) and the sports level represented by the concerned group (level of competitions). Such information makes it easier for the reader to transfer the results to their training sessions.

Please correct the description of the columns in Table 6 (which results refer to "RTG" and which to "HIPT")

Experimental design

Line 246 - To standardize the test results, the protocols of warm-ups performed before control tests (before and after "HIPT") should be described in detail
I recommend supplementing this scheme in describing all research tests performed in the study, provided they were present.

Validity of the findings

Line 288 - 292 - I propose to move these two sentences to the conclusion section.
In the Discussion section - we refer to the research results obtained by other authors on our own and propose (according to our knowledge) interpretations, e.g. what caused differences or similarities.

It is always based on the available literature included in the introduction or the discussion.


Conclusions section - Move part of this paragraph to the Discussion section. Then edit your findings and suggestions as previously instructed.

In a work of such a practical nature, there is no information about:
Research limitations
Suggestions for further research
and practical use both in football and other sports

Additional comments

The work is almost ready to be made public; it contains exciting and practical solutions that can support the training process at every level of the game, perhaps even in senior teams. The HIPT-based intervention itself is an accepted and effective concept widely used. Each new protocol can bring a few more important details to working on another coaching ground in different places worldwide.
Please consider my suggestions carefully and make corrections quickly.
I wish the authors many scientific and sports successes in their future careers.

Good luck

PhD Michał Nowak
educationsupport@icloud.com
University of Jan Długosz in Częstochowa
COLLEGIUM MEDICUM name WŁADYSŁAW BIEGANSKI
Department of Physical Culture Sciences:

·

Basic reporting

The purpose of this study was to examine the effects of high-intensity plyometric training (HIPT) on some parameters in elite football players in the U17. I believe that this experimental procedure includes a lot of insightful information with useful practical suggestions for improving performance in athletes. However, I noticed a few notes that needed clarification. The most important remark is that the part of the discussion must be more detailed

Experimental design

INTRODUCTION
Point 4: Try to delete the common information in the two paragraph of the introductory part.
Point 5: Line 66-70 : Please rephrase this sentence.
Point 5: Line 70-72 : If add an appropriate reference.
Point 6: It is better to present the problematic of your study as clearly as possible.
MATERIALS AND METHODS
Point 7 : Specify the period of the intervention in Study Design part.
Point 8 : Mention and detail the pre-season preparation period if possible.
Point 9: Add more details regarding the study population (nationality, which division, etc.) in participants part.
Point 7: How did you guarantee that both groups followed the same volume of training? Add this information in experimental approach part.
Point 8: On what basis did you choose the types of exercises, volume, intensity, etc., of the plyometric training program? Add this information and the appropriate reference to the experimental approach.

DISCUSSION
Point 9 : Line 294-297 : Missing reference.
Point 10 : Line 306-520 : Reduce the number of cited references
Point 11 : Lack of scientific explanation for improvement in sprint performance after the pliometric training program.
Point 12 : Line 332-355 : Reduce the number of cited references
Point 13 : Line 356-361 : Justify the interest of adding this paragraph.
Point 14 : Lack of scientific explanation for improvement in long jump performance after the pliometric training program.
Point 15 : Too many references in Triple Hop (Right and Left) Performance part.
Point 16 : Lack of scientific explanation for improvement in triple hop (Right and Left) performance after the pliometric training program.
Point 17 : Too many references in 505 COD Performance part.
Point 18 : Lack of scientific explanation for improvement in 505 COD Performance after the pliometric training program.

CONCLUSION
Point 19 : It is best to add practical recommendations at the end of the concluding part in order to protect participants from injury, especially during plyometric training.
Point 20 : if possible add part of study limits.

Validity of the findings

Nothing to add.

Additional comments

TITLE
Point 1: It is better to add the specialty of the study population to the title
ABSTRACT
Point 2: Line 24 : This abbreviation is unclear ‘’RT’’.
Point 3: more detail about the results, it is better to reduce the statistical part.

---

## Round 0.2 · Major Revisions

Dear authors,
The article needs some major revisions. Indeed, as pointed out by reviewer#1, the presentation of the article needs some work. For example:

• 505 change-of-direction test is better explained, but there are some parts that needs to be clarified (e.g., the authors use A and C point but it is not clear to what they are referring to, or they added Figure 2 but it is not called-out or introduced in the text);
• Line 322-324, the explanation of the acceptable thresholds for kurtosis and skewness is not clearly written;
• The tables included in the results section should be introduced in the text, and the explanations included after tables 6-12 seem to be more appropriate to be moved to the text of the manuscript instead of using them as captions;
• Conclusion and suggestions chapter. This chapter could be adjusted by focusing more on the results of the study in the context of the literature and limiting the citations in the first part. Then, talking about future directions. The limitations, on the other hand, could be moved to the end of the discussion section. Finally, the title can be changed to "Conclusions".

Please revise the article considering the reviewer#1 comment and provide a point-by-point rebuttal addressing every point of this round of revision and those unaddressed in the previous round.
Carlo

·

Basic reporting

Good morning.
Undertaking the analysis of the work once again, I propose the following changes to improve the quality and readability of the work:

First of all, the introduction continues with opinions such as: "In addition, plyometric training, which can be performed by soccer players to increase explosive strength, is one of the best training practices."
This should be reworded so that the term "one of the best practices" is too long - not everyone is the same, and they do not always react identically to a training intervention.

still the same

another quote, "Research shows that plyometric training is one of the most effective specific training methods that can directly impact match performance and develop strength/power (William and Kirubakar, 20, 18). "
For the following:
In the opinion of researchers (XXX, YYYY), plyometric training is an effective training method that improves such parameters.

Once again, I would like to point out that in the introduction, researchers describe that the athlete's body composition is an important factor. We all agree with this, but in this study there are no results related to changes in parameters, e.g. muscle mass or fat tissue in relation to the tested groups. There is no search for cause and effect relationships related to test results.

This paragraph should be removed from the introduction because it does not contribute anything to the quality of the work

The introduction does not provide exhaustive information about the impact of high-intensity plyometry training on anaerobic parameters.

The information contained in line number 140 - regarding "Preseason training" should be moved to the introduction section - it has nothing to do with the material and work methods

There are a lot of tables in my work. For clarity, I recommend making the following modifications.
Table 1 - not doubling the information contained in the table in the text below is a big mistake.

Table 2. The use of "Football training - FT abbreviations, i.e. unification of the whole.

convert table 3 into a chart (this will reduce the number of tables and will be more readable compared to the text in the work)

Tables in work results:
The results section should not consist only of tables. It is not known whether the text under the tables is a table caption or text. The tables need to be improved

Experimental design

Further, from my knowledge, the quasi-experiment is most often used when there is no control group. There is a control group in this study, so it's probably not that type.

The results section should not consist only of tables. It is not known whether the text under the tables is a table caption or text. The tables need to be improved

Validity of the findings

Preparing a chapter called discussion involves comparing your own results with those of other authors in order to confirm or deny their effectiveness. Based on the information contained in the introduction chapter and additionally in the discussion chapter.

The discussion doesn't have to be long. The most important thing is the quality and compatibility of the information with the results provided by the authors.

Discussion section - Should be improved.

Additional comments

The authors did not make significant changes to their manuscript.
The research they have done is important, but its presentation leaves much to be desired.

Taking into account that there is another approach to the review, I reject this manuscript.

·

Basic reporting

The purpose of this study was to examine the effects of high-intensity plyometric training (HIPT) on some parameters in elite football players in the U17.
I believe that this experimental procedure includes a lot of insightful information with useful practical suggestions for improving performance in athletes.
The authors answered all questions and suggestions well and improved all parts of the manuscript.

Experimental design

All answers are accepted.

Validity of the findings

All answers are accepted.

Additional comments

All answers are accepted.

---

## Round 0.3 · Minor Revisions

Dear authors,

Most of the comments have been addressed; however, the comment regarding the table and results was not handled.

In particular, it is true that the authors should not repeat the information in the text and tables and that the table should be self-explanatory; however, there must be a call-out for each table in the results written section and some of the table captions are reported in an uncommon way, which is not appropriate for a table caption.

For example, what is written in Table 4, lines 4-10, should be adapted for a table caption because it is written how it would be written in the text of the manuscript.

Additionally, in the PDF, there is a mismatch between the numbers of the tables, their order, and the information.

So, I kindly ask the authors to please address these points.
Regards,
Carlo

·

Basic reporting

The corrections have been incorporated - I have no further suggestions

Experimental design

The corrections have been incorporated - I have no further suggestions

Validity of the findings

The corrections have been incorporated - I have no further suggestions

---

## Round 0.4 · Minor Revisions

Dear authors,

The issue related to the presentation of the result still needs to be handled.

In the results section, there must be a written part in which you introduce the tables. For example:

Results
Participants' characteristics are reported in Table XX.
As shown in Table xxx, there were no significant differences between...

This is not done in the pdf the authors uploaded since the result section, lines 325–342, is just a list of the tables.

Additionally, the information reported in the captions of tables 4 to 10 is not appropriate as a table caption. Indeed, the parts that start with “When...” of the tables are not what should be in a caption but what should be in the text of the manuscript in the result section.

This must be addressed.
Regards,
Carlo

---

## Round 0.5 · accepted · Accept

The article is now suitable for publication in PeerJ.